# OpenReview forum: "Deterministic Discrete Denoising"
_ICML.cc/2026/Conference — Submitted to ICML 2026_

### Official Review · Reviewer_HYCN · 2026-02-28

**Soundness:** 2
**Presentation:** 2
**Significance:** 3
**Originality:** 2
**Overall Recommendation:** 3
**Confidence:** 4

**Summary:**

This paper addresses the core challenge of discrete-state diffusion models—the implementation of deterministic denoising in discrete state spaces—and proposes a deterministic denoising method based on a variant of the herding algorithm. This method can directly replace the original stochastic denoising process with the advantages of no model retraining and no need for embedding discrete data into continuous spaces, achieving a drop-in replacement effect. It thus fills the research gap of the lack of deterministic denoising methods for discrete uniform diffusion models. The paper theoretically proves the boundedness of the weight vector and the discrepancy convergence rate, and verifies the effectiveness of the algorithm on three core tasks: text generation, image generation, and combinatorial optimization. Moreover, the experiments exhibit good reproducibility (needing only about 20 lines of code modification). The overall research has a clear logic and distinct innovations, providing certain reference value for the relevant research on discrete-state diffusion models. However, there are significant deficiencies in several aspects of the study, and the persuasiveness of some core conclusions and the practical deployment capability of the algorithm need to be further improved.

**Compliance With Llm Reviewing Policy:**

Affirmed.

**Final Justification:**

I still believe it is necessary to further validate the generalization of the proposed method on more complex datasets, which remains a key concern of mine. Accordingly, I choose to maintain my current review score.

**Key Questions For Authors:**

(1) Does the impact of the marginal parameter $\delta$ for delayed switching on the algorithm's performance have a threshold effect? That is, will the algorithm's performance experience a cliff-like drop when $\delta$ exceeds a certain critical value? Is it possible to design an adaptive selection strategy for $\delta$ to adapt to different application scenarios?

(2) The weight vector in the paper is updated by simple accumulation at each time step. Have you considered introducing a temporal decay factor for weight update? Can this improvement further enhance the convergence effect of the algorithm?

(3) For high-dimensional discrete spaces (e.g., large language models with a vocabulary size of over 100,000), the dimension of the weight vector increases linearly with the vocabulary size, and the direct application of the algorithm is prone to low efficiency. Are there any specific performance improvement schemes for this problem?

(4) What are the core differences in dynamic characteristics between the modified time-dependent herding algorithm and the original herding algorithm? How do these differences specifically affect the convergence rate and sampling effect of the algorithm?

**Limitations:**

See the **Weaknesses** section and the **Key Questions For Authors** section above.

**Strengths And Weaknesses:**

**Strengths:**

(1) The herding algorithm is modified with time dependence and introduced into the reverse denoising process of discrete diffusion, for the first time proposing a deterministic denoising method suitable for discrete uniform diffusion models and filling the gap in existing research. Meanwhile, it is confirmed that the deterministic reverse process, which has been well established in continuous diffusion, is also effective in discrete state spaces.

(2) The paper rigorously proves the boundedness of the weight vector and the $\mathcal{O}(T^{-1}) $ convergence rate of the discrepancy between the empirical sample distribution and the averaged probability vector from a theoretical perspective, providing a theoretical foundation for the algorithm's effectiveness. Furthermore, this convergence rate is significantly superior to the $\mathcal{O}(T^{-1/2}) $ rate of stochastic sampling, which explains the efficiency advantage of the algorithm theoretically.

(3) The algorithm is implemented as a drop-in replacement module that requires no modification to the network structure of existing discrete diffusion models or their retraining. Its deployment only needs about 20 lines of code modification in the reverse denoising process. It exhibits good adaptability in both the UDLM and DIFUSCO frameworks, effectively reducing the engineering cost of algorithm deployment.

(4) The experiments cover four sub-tasks: character-level/word-level text generation, discretized image generation, and TSP combinatorial optimization, involving two core application scenarios of discrete diffusion (generative modeling and combinatorial optimization). The algorithm significantly outperforms the original stochastic denoising method on most evaluation metrics, verifying the generalizability of the algorithm.

(5) The paper analyzes discrete diffusion models from a novel perspective of nonlinear dynamics and deterministic sampling, combining the weakly chaotic/piecewise isometric dynamic characteristics of the herding algorithm with the state transition process of discrete diffusion. It thus provides new ideas and perspectives for the subsequent research on discrete-state diffusion models.

**Weaknesses:**

(1) Computational complexity analysis is an essential component of such algorithm research, but this paper does not conduct any time or space complexity analysis of the proposed algorithm, making it impossible to clarify the algorithm's computing power and efficiency performance in high-dimensional discrete spaces.

(2) The marginal parameter $\delta$ for delayed switching is the core hyperparameter determining the algorithm's performance. Its value is highly dependent on factors such as model architecture, dataset type, vocabulary size $K$, and number of inference steps $T$, with no clear setting criteria, which greatly increases the practical application cost of the algorithm. Designing an adaptive selection strategy for $\delta$ or providing value guidelines for different application scenarios would significantly enhance the practical application potential of this research.

(3) When comparing with the state-of-the-art algorithm DNDM, the authors explicitly state that the experimental settings are inconsistent, and no fair comparative experiments are conducted via control variables. The performance advantage of the proposed algorithm is only demonstrated by the numerical difference in perplexity (PPL), which results in insufficient persuasiveness of this conclusion.

(4) The algorithm's validation is only completed on a limited set of datasets, and no experiments are conducted on typical discrete diffusion datasets such as genome sequences, chemical molecular structures, and graph data, making it impossible to fully verify the algorithm's generalizability on different types of discrete data.

(5) All experiments are carried out in small and medium-scale discrete spaces, with no validation of the algorithm's performance and efficiency in large-scale discrete spaces (e.g., large language models with massive vocabularies, high-resolution discretized images), which makes it impossible to clarify the algorithm's performance and application potential in practical industrial scenarios. In addition, ablation studies are not designed in the research, so the independent contributions of each core module of the algorithm cannot be clarified.

---

> ### Author Rebuttal · Authors · 2026-03-31
>
> Thank you for your insightful review. We appreciate your detailed assessment of the novelty, implementability, and broad empirical evaluation of our work. Below we respond point by point.
> ### Answers to Key Questions
> **(1) Impact of $\delta$.**
> Our experiments suggest that the dependence on $\delta$ is gradual rather than cliff-like. While our manuscript reports near-best results, our proposed method improves over probabilistic sampling for a broad range of $\delta$ values. For example, on LM1B, we observed that any $\delta \in [0, 0.5]$ improved generative PPL for both $T=128$ and $T=1024$ (see the table in response to Reviewer B5g3). We agree that an adaptive strategy would further strengthen practical applicability. At present, we observe that larger $\delta$ tends to work better for smaller $T$. We will include these results in the revision.
>
> **(2) Decayed weight vector.**
> Thank you for this insightful suggestion. Introducing temporal decay interpolates between our proposed algorithm and the greedy stateless argmax update, corresponding to no decay and extremely fast decay, respectively. The former reduces accumulated discrepancy, while the latter reduces the discrepancy at each time step. An intermediate decay may further improve performance. However, in our paper, we intentionally kept the method simple, since our main goal is to establish deterministic discrete-state denoising as a promising approach. We view decayed variants as one of several promising future directions.
>
> **(3) High-dimensional discrete spaces.**
> We agree that scalability to very large discrete spaces is important in practice. To clarify, the time and space complexities of both the proposed and the original algorithms scale linearly in $K$. Excluding the prediction model, the time complexity of both methods is $O(TKL)$, reducible to $O(TK)$ by parallelizing across the $L$ tokens, while the space complexity remains $O(KL)$. As you pointed out, the proposed method requires an additional weight matrix of size $K \times L$ and its manipulation, so although the asymptotic order is unchanged, the actual memory usage and wall-clock time increase. In practice, however, the wall-clock cost is dominated by the prediction model. On the same machine with an NVIDIA A100 GPU, generating 1024 LM1B samples with $T=1024$ took 1121 seconds with the proposed method and 1088 seconds with the original Gumbel-max sampling, indicating that the additional overhead is small. To further reduce the time complexity, approximate argmax algorithms are a natural direction, since an exact argmax is not required for the $O(T^{-1})$ discrepancy bound. Moreover, our proposed method can reach comparable quality with fewer NFEs, effectively reducing the dominant prediction model cost.
>
> **(4) Difference in dynamic characteristics.**
> We needed a time-dependent variant to apply herding to the reverse process, where the target probability vector varies over time. If the target probability vector is fixed, the dynamics essentially reduce to the original herding system. The time-dependent version inherits the dynamical characteristics of the original herding system, including the $O(T^{-1})$ discrepancy and the piecewise isometry.
> ### Clarifications on Weaknesses
> **(1)(2) Computational complexity analysis / Hyperparameter tuning.**
> These points are clarified in our responses to Key Questions (3) and (1), respectively.
>
> **(3) Comparison with DNDM.**
> Our main conclusion on text8 is the comparison between the proposed deterministic denoising algorithm and the original probabilistic sampling process. Therefore, the comparison with DNDM is not central to our main claim. We agree that our performance comparison should be presented more carefully, and we will revise the exposition accordingly.
>
> **(4) Limited set of datasets.**
> We agree that extending the range of datasets is important. Since our method improves only the reverse denoising process and does not modify the prediction model itself, we need sample-quality metrics to evaluate generated outputs. Such metrics are particularly well established for text and image generation, which is why we focused on these domains. We also included combinatorial optimization, where samples can be evaluated directly through objective values. Based on these results, applying our method to domains such as biological sequences, molecules, and graphs is an important future direction.
>
> **(5) Large-scale / ablation study.**
> It is true that the vocabulary size in our experiments is limited to around 50,000. However, as answered above for Key Question (3), the additional overhead is small relative to the prediction model cost. As for ablation, our contribution is a single drop-in algorithm, and the comparison presented in the paper between the original and derandomized sampling algorithms with the same model isolates the contribution of the proposed method. The dependence on $\delta$ is addressed separately in Key Question (1).

---

> > ### Author Rebuttal · Reviewer_HYCN · 2026-04-03
> >
> > Thank the authors for their response. However, I still believe it is necessary to further validate the generalization of the proposed method on more complex datasets, which remains a key concern of mine. Accordingly, I choose to maintain my current review score.

---

> > > ### Author Response · Authors · 2026-04-04
> > >
> > > Thank you very much for your follow-up comment clarifying your remaining concern. We agree that validation on more complex discrete datasets would further strengthen the paper. As shown in our rebuttal to Reviewer TrUM, we additionally evaluated our method on the OpenWebText dataset, which is substantially more complex than LM1B in terms of domain diversity, document-level structure, and textual noisiness. We confirmed that the improvement generalizes to OpenWebText: at $T=1024$, the generative PPL decreases from 72.85 with probabilistic denoising to 40.84 with the proposed herding-based denoising. This experiment also enables comparison with other methods under the same training conditions; e.g., the PPL values at $T=1024$ are 105.64 for MDLM and 75.02 for Duo (Deschenaux et al., arXiv:2602.21185). We would like to include these additional empirical results in the revision, and we hope this helps further address your concern.

---

### Official Review · Reviewer_GBXx · 2026-03-11

**Soundness:** 1
**Presentation:** 3
**Significance:** 2
**Originality:** 3
**Overall Recommendation:** 3
**Confidence:** 4

**Summary:**

The authors propose an approach for deterministic sampling from a discrete diffusion based on uniform noising. The authors base their approach on the herding algorithm for sampling a sequence of discrete variables whose features converge to the prespecified target vector. The authors propose an adaptation of this approach for using in sampling from the reverse process of discrete diffusion. The authors tested their plug–in sampling approach, taking trained discrete diffusion models on character-level generation, text-level generation, and image generation. They also test it for diffusion-based combinatorial optimisation.

**Compliance With Llm Reviewing Policy:**

Affirmed.

**Final Justification:**

The authors partially addressed my concern; however, proper theoretical justification of the methods is still missing and text evaluations are not complete.

**Key Questions For Authors:**

- Is it possible to provide strict theoretical justification for this sampling scheme?

- Why is DNDM compared with DDIM? DNDM derandomizes only transition times, not transitions.

**Limitations:**

Yes

**Strengths And Weaknesses:**

Strengths:
- The proposed method and motivation for it are presented well in a clear manner.
- The proposed method is easy to implement and can be used with any pretrained discrete diffusion based on uniform noising.
- The authors consider the important problem of derandomization of sampling from the reverse process of discrete diffusion and show how it can be linked to the herding algorithm.
- Proposed sampling shows improvement on CIFAR setup. Although this setup is not practically meaningful, taking into account typical FID values for discrete diffusion.

Weaknesses:
- Proposition 2 is basically about the fact that averaged samples over trajectories converge to the average probability vectors over trajectories. However, this does not provide any information on how close the samples are to the true marginal distributions or how close the final samples are to the model distribution.
- There are no theoretical guarantees on the proposed sampling algorithm (taking into account that Proposition 2 does not provide information on the correctness of the proposed sampling algorithm). Without it, the claims about
- Authors do not try to theoretically analyze the version with a margin used in practise.
- Results on Text8 and LM1B datasets show better perplexity at the cost of lower entropy. However, this is well know relation, that lowering the entropy of the distribution leads to better generative perplexity. Thus, it is hard to deduce whether there are any improvements on these setups.
- Overall, the algorithm does not have any theoretical justification for derandomization in sampling. As a heuristic approach, it is not clear whether it improves the quality of texts, while improvement on images is more demonstrative than practical.

Minor weaknesses:
- Comparison with DNDM is only in text; the results are missing in the corresponding figures and tables.

---

> ### Author Rebuttal · Authors · 2026-03-31
>
> Thank you for your careful review. We appreciate your positive assessment of the clarity, implementability, and originality of the proposed method. Below we respond point by point.
> ### Answers to Key Questions
> **(1) Strict theoretical justification.**
> For a categorical distribution with the probability vector as the feature target, the original herding system yields samples from the same distribution efficiently. However, we would like to clarify that the original herding system in general is not an exact sampling method, but rather a deterministic approximation with prominent $O(T^{-1})$ discrepancy convergence (Welling, ICML 2009). Even for the original herding system, the induced sample distribution is theoretically difficult to characterize. Subsequent work (Harvey and Samadi, COLT 2014) conjectured, based on both numerical and theoretical analyses, that the herding algorithm yields samples from the maximum-entropy distribution subject to the target feature constraints. This is why feature design is important in herding. In this regard, we propose to design such a feature for the reverse process of discrete diffusion models, and we employ the averaged sample vector as a starting point. Propositions 1 and 2 theoretically support this design by establishing boundedness and the inherited $O(T^{-1})$ discrepancy bound. However, this feature design is not unique, as you noted, and it is possible that other features, such as decayed variants suggested by Reviewer HYCN, may perform better. We agree that the theoretical understanding of the final sample distribution remains limited, and a more complete characterization is an important direction for future work.
>
> **(2) DNDM and DDIM.**
> Our understanding is that the DNDM paper (Chen et al., NeurIPS 2024) explicitly positions the method in relation to DDIM (see their Table 1) and describes it as “enabling deterministic evolution” and “the reverse sampling becomes deterministic” upon using their Eq. (9). However, we agree that our performance comparison with DNDM should be presented more carefully, and we will revise the exposition accordingly.
> ### Clarifications on Weaknesses
> **(1)(2) Theoretical guarantees on final sample distribution.**
> These points are clarified in our response to Key Question (1).
>
> **(3) Theoretical analysis with margin.**
> We would like to clarify that Propositions 1 and 2 are mathematically proved for the version with the margin parameter $\delta$, since the proof of Proposition 1 incorporates $\delta$ into the lower bound.
>
> **(4) Text results.**
> We agree that generative PPL should not be interpreted in isolation, especially when entropy also changes. However, it is also true that the improvement in generative PPL cannot be attributed solely to the decrease in unigram entropy. This is why we reported not only generative PPL and entropy, but also MAUVE (Pillutla et al., NeurIPS 2021), which measures the closeness between sample distribution and real-data distribution based on the divergence frontiers. In particular, for LM1B, the proposed method improves both generative PPL and MAUVE, so the improvement is not assessed from perplexity alone.
>
> **(5) Quality evaluation.**
> Evaluating sample quality is inherently difficult in generative modeling. For faithful comparison with prior work, we followed the same evaluation metrics as the UDLM paper (Schiff et al., ICLR 2025). In text, we report generative PPL and entropy as in the prior work, and additionally report MAUVE as discussed above. In images, we employ the same uniform-noising discrete diffusion model as in the prior work. We also included combinatorial optimization, where samples can be evaluated directly through objective values.

---

> > ### Author Rebuttal · Reviewer_GBXx · 2026-04-03
> >
> > Thank you for the provided answers. My concerns are partially addressed. Remaining concerns are:
> >
> > **Justification for the algorithm.** The problem is that the provided algorithm is essentially not theoretically justified, but it is written like there is such justification. The only convergence property from Proposition 2 is basically that the average sampled distribution over all times $t$ converges to the average true distribution over all times $t$. For instance, let’s say that the true distribution of a two-dimensional vector is [1, 0] for 0 <= t < 0.5 and [0, 1] for 0.5 <= t <= 1.0. Then, on time average it is [0.5, 0.5]. Then let’s say that the sampled distribution fully misplaced probability mass and has values [0, 1] for 0 <= t < 0.5 and [1, 0] for 0.5 <= t <= 1.0, then on time average it is [0.5, 0.5]. Hence, in this case, the simulated trajectory is completely wrong both in dynamics and at a final moment $t=0$, while on time average it is perfect. Hence, such features cannot distinguish between perfectly simulated dynamics and ideally wrong simulated dynamics.
> >
> > **DNDM.** It is «deterministic» in the sense that times of transition are deterministic, not trajectories.
> >
> > **Empirical evaluation on texts.** Without having similar entropy, generative perplexities are not comparable since there is a strict correlation between reducing entropy and generative perplexity. Mauve is a valid metric, but it is reported only for one reproduced baseline and only for one setup. Also, there is no value for real data, so it is hard to understand how the reported MAUVE compares with real data.

---

> > > ### Author Response · Authors · 2026-04-04
> > >
> > > Thank you very much for your follow-up comment. We hope the following explanations further address your concerns.
> > >
> > > **Justification for the algorithm.** Let us explain this point based on your example, which is very helpful for clarification. First, we agree that the statement of Proposition 2 itself only guarantees the $O(T^{-1})$ convergence of the discrepancy averaged over the full interval, and therefore does not distinguish a wrong trajectory satisfying the correct full-time average, as you explain. However, Proposition 1 states that the discrepancy accumulated over $\tau \le t \le 1$ at each intermediate $\tau$ is bounded, and the herding dynamics greedily reduce this discrepancy at each step. Therefore, in your example, the sample distribution over the interval $0.5 \le t \le 1.0$ is driven to be close to $[0,1]$. After that, the sample distribution over the full interval $0 \le t \le 1.0$ is also driven toward the time average $[0.5,0.5]$. This implies that the sample distribution over the subinterval $0 \le t \le 0.5$ must be close to $[1,0]$, as expected. This is what we mean by describing the proposed dynamics as following the reverse Markov chain faithfully. Of course, this argument does not provide a rigorous justification for the terminal sample distribution. Your comment made us realize that our explanation in the manuscript was insufficient in two respects: (1) how minimizing the discrepancy at each step supports faithful reverse dynamics, and (2) that the terminal sample distribution is not theoretically justified. We will revise the exposition accordingly.
> > >
> > > **DNDM.** We agree that the entire DNDM algorithm is not deterministic. Our discussion of DNDM is not central to our study. Your comment also helps clarify that transitions are derandomized in our method. We will revise the exposition accordingly.
> > >
> > > **Empirical evaluation on texts.** We agree that there is a trade-off between generative PPL and joint entropy. At the same time, generative PPL together with unigram entropy is a standard evaluation setup widely used in prior work on discrete diffusion language models, so following this convention is informative for comparison. We also report MAUVE as a complementary metric to address such concerns. However, published MAUVE scores are unavailable for prior discrete diffusion language models, which is why our MAUVE comparison is limited to a single reproduced baseline. Still, this is sufficient to support our main conclusion, namely the improvement over the original probabilistic denoising, rather than to establish superiority over other prior models. We will try to include additional MAUVE results in the revision. The MAUVE score of real data is (close to) 1 by definition, and the score of herding-based denoising is closer to 1 than that of probabilistic denoising. We admit that all of these text metrics have inevitable deficiencies as proxies for human evaluation. More broadly, since the scope of this work is not limited to text, we assessed the proposed method from multiple perspectives by also evaluating it on image generation and combinatorial optimization.

---

### Official Review · Reviewer_TrUM · 2026-03-11

**Soundness:** 2
**Presentation:** 4
**Significance:** 3
**Originality:** 3
**Overall Recommendation:** 5
**Confidence:** 3

**Summary:**

This paper proposes a deterministic sampling method for discrete diffusion models based on herding. The idea is to replace the usual stochastic reverse sampling with a deterministic update rule. The method is applied mainly to uniform discrete diffusion, and the experiments show improvements across text, categorical image generation, and diffusion-based combinatorial optimization. The main contribution is showing that deterministic herding can be effective in discrete diffusion.

**Compliance With Llm Reviewing Policy:**

Affirmed.

**Final Justification:**

I am still unfortunately unconvinced by the motivation for deterministic sampling. In their final response, the authors state that "taking larger time step sizes means that, in each reverse denoising step, the state transition probabilities become larger". I may be mistaken, but I do not see why this point depends on deterministic rather than stochastic sampling. My understanding in the SDE vs ODE setting is that deterministic sampling can help because the Brownian motion requires a finer discretization than the finite variation part of the process. I have not been convinced that an analogous argument applies in the discrete setting. As a result, I am currently only able to view the method as a neat heuristic that appears to work well in practice, rather than as something whose underlying motivation is fully justified. For that reason, I do not feel inclined to champion the paper.

At the same time, I enjoyed reading the paper, and I believe it could spark interest and generate further ideas for other readers if accepted. It has made me think more carefully about related topics. I have therefore raised my score to 5, and I would be happy to see the paper accepted.

**Key Questions For Authors:**

1) What is the main reason to prefer deterministic sampling in the discrete setting?
In continuous diffusion, deterministic samplers are often motivated by the ability to take larger steps with fewer NFEs, but in discrete diffusion that motivation seems less clear to me. Could the authors clarify the setting in which deterministic reverse sampling should be expected to help, beyond the empirical observation that it does here?

2) Can the authors better connect the discrepancy result to terminal sample quality?
The theory argues that the herding trajectory stays close to the reverse transition probabilities on average, but it is not clear to me why this should imply better final samples, which is the quantity that ultimately matters. Do the authors have either a stronger theoretical connection or empirical evidence that lower discrepancy correlates with improved terminal quality?

3) What is the intuition for why herding, in particular, should help in discrete diffusion?
The empirical improvements are strong, but I found the mechanism unclear. Is there an intuition for why herding should outperform standard stochastic sampling?

4) Could the authors report the Gen. PPL and entropy of the real LM1B dataset under the same evaluator?
This would help interpret the LM1B numbers.

**Limitations:**

Yes

**Strengths And Weaknesses:**

The paper’s main strength is that it proposes a simple deterministic sampling rule for discrete diffusion that can be used without retraining, and the empirical results are strong across several domains.

On presentation, the paper is generally clear and easy to follow, and the method itself is simple to understand.
On significance, the problem is important and a drop-in sampler that improves results without retraining is practically useful.
On originality, I think the contribution is meaningful. The ingredients are not all new, but the combination is novel enough.

My main concern is on soundness, since I found the empirical case to be much stronger than the theoretical one. While the experiments are broad and convincing overall, I was not fully persuaded by the paper’s explanation for why deterministic sampling should help in discrete diffusion in the first place. Secondly, the discrepancy argument (i.e., the trajectories are close to the reverse transition probabilities on average) does not clearly connect to terminal sample quality. I also think the paper gives limited intuition for why herding should improve performance beyond the empirical observation that it does. Finally, some of the LM1B results seem surprisingly strong, and I would have liked more discussion of how to interpret those numbers.

---

> ### Author Rebuttal · Authors · 2026-03-31
>
> Thank you for your careful and constructive review. We appreciate your positive assessment of the clarity, originality, and practical usefulness of our contribution. Below we respond point by point.
> ### Answers to Key Questions
> **(1) Motivation for deterministic sampling.**
> The main reason to prefer deterministic sampling in our proposal is the same as in the continuous case: improved efficiency and sample quality, i.e., achieving better quality for the same NFEs ($T$), and comparable quality with fewer NFEs, as demonstrated in our experiments. Beyond this empirical advantage, the deterministic reverse process also enables novel applications that are difficult to formulate in the fully stochastic setting. For example, one can imagine applications such as discrete morphing by interpolating between the initial states of two generated samples, analogous to what deterministic samplers have enabled in continuous diffusion models.
>
> **(2)(3) Terminal sample distribution / Why herding.**
> The success of deterministic samplers in continuous diffusion models is a strong motivation for derandomization in discrete state spaces. Our intuition for the promising experimental results is that derandomization enhances iterative refinement by following the reverse Markov chain more faithfully. However, a theoretical justification for the terminal sample distribution is not straightforward. The original herding system in general is not an exact sampling method, but rather a deterministic approximation with prominent $O(T^{-1})$ discrepancy. Since the approximation quality of herding depends on the chosen target features, their design is important. In this regard, we propose to design such a feature for the reverse process of discrete diffusion models, and we employ the averaged sample vector as a starting point. Propositions 1 and 2 theoretically support this design by establishing boundedness and the inherited $O(T^{-1})$ discrepancy bound. However, this feature design is not unique, and both better feature design and a characterization of the terminal sample distribution remain important directions for future work.
>
> **(4) Real-data statistics on LM1B.**
> Thank you for this helpful suggestion. We performed evaluation of generative PPL and unigram entropy on the LM1B dataset using the same evaluators. The results are: Gen. PPL = 47.37 and entropy = 7.51. The obtained PPL value is almost consistent with the value 44.752 reported for the 762M model in the GPT-2 paper (Table 3 in Radford et al., OpenAI, 2019). Our reported PPL value 46.71 for UDLM+herding is comparable to the real-data value, while the entropy of the UDLM samples is lower than that of the dataset. Of course, this result does not necessarily mean that the generated samples are real-data level, as GPT-2 large model is just an incomplete proxy for human assessment. Therefore, our claim remains that herding-based denoising significantly improves Gen. PPL under the GPT-2 large evaluator.
> ### Clarifications on Weaknesses
> The points raised in the review are mostly clarified in our responses to the Key Questions above. As for the strong LM1B results, our interpretation is that derandomization enhances iterative refinement by following the reverse Markov chain more faithfully, as can be seen in Table 2, where the PPL improves as the number of inference steps $T$ increases. We recently confirmed the same tendency on another text dataset, OpenWebText, which is larger than LM1B. After the ICML submission, we obtained a UDLM model trained on OpenWebText, which facilitates comparison with other methods under the same training conditions. For reference, we include below supplementary OpenWebText results. The table shows that the PPL of UDLM with herding (fixed $\delta=0.01$) continues to improve as $T$ increases, and that the combination of UDLM and herding outperforms both MDLM and Duo. Adjusting $\delta$ depending on $T$ further improves performance. We also note that the MDLM and Duo results (Table 12 in Deschenaux et al., arXiv:2602.21185) include further improvements using distillation and nucleus sampling, and these techniques could in principle also be combined with herding.
>
> **Supplementary table.** Gen. PPL (entropy) of generated samples from models on OpenWebText.
>
> |              |        $T=32$ |        $T=64$ |       $T=128$ |       $T=256$ |       $T=512$ |      $T=1024$ |
> | ------------ | ------------: | ------------: | ------------: | ------------: | ------------: | ------------: |
> | MDLM         | 194.09 (5.74) | 141.67 (5.69) | 120.95 (5.67) | 111.85 (5.65) | 107.89 (5.64) | 105.64 (5.64) |
> | Duo          |  96.76 (5.57) |  86.01 (5.56) |  79.97 (5.55) |  78.46 (5.53) |  76.93 (5.54) |  75.02 (5.53) |
> | UDLM         |  83.71 (7.24) |  77.07 (7.31) |  73.71 (7.35) |  72.83 (7.39) |  73.14 (7.44) |  72.85 (7.43) |
> | UDLM herding |  92.09 (6.76) |  60.15 (6.81) |  51.25 (7.05) |  49.44 (7.30) |  42.52 (7.30) |  40.84 (7.33) |

---

> > ### Author Rebuttal · Reviewer_TrUM · 2026-04-02
> >
> > Q1a: Would you mind explaining how deterministic sampling improves efficiency? In the continuous case it allows larger step sizes, but I do not see how this carries over to the discrete case.
> >
> > Q1b: Also am I correct in understanding that the claim on the benefits of deterministic sampling on sample quality comes mainly from empirical evidence, but there is no intuition for why this occurs? (I am satisfied if this is the case, but I just want to make sure I understand.)
> >
> > Q2/3: Would you mind explaining how Propositions 1 and 2 and the inherited $O(T^{-1})$ discrepancy bound relate to improved final sample quality?
> >
> > Q4: Thank you for your response.

---

> > > ### Author Response · Authors · 2026-04-02
> > >
> > > Thank you very much for your insightful follow-up questions, which help clarify the central points.
> > >
> > > Q1a: In the continuous case, taking larger time step sizes generally increases the length of a single move in the state space, as you point out. In the discrete case, the corresponding effect appears in the Markov transition probabilities: taking larger time step sizes means that, in each reverse denoising step, the state transition probabilities become larger. Our deterministic sampling follows these transition probabilities more faithfully, and empirically maintains sample quality even with larger time step sizes, or equivalently fewer NFEs ($T$).
> > >
> > > Q1b: You are right that our claim about improved quality and efficiency is based mainly on empirical evidence. As for intuition, this may depend on the level of intuition you have in mind, but, as noted in our previous rebuttal, our main intuition is that the deterministic $O(T^{-1})$ discrepancy, compared with the probabilistic $O(T^{-1/2})$ discrepancy, allows the reverse process to follow the reverse Markov chain more faithfully, thereby improving iterative refinement in practice. Our study is also motivated by the broader intuition that careful derandomization can improve sampling. While probabilistic sampling is simple and admits cleaner theoretical guarantees, this does not mean that sampling must be probabilistic. Our work extends this idea to discrete-state diffusion models.
> > >
> > > Q2/3: Beyond the theoretically established discrepancy bound, we currently have only the intuitive explanation described above. Establishing a theoretical connection to final sample quality should be an important direction for future work, given the empirically promising results in our paper.
> > >
> > > Q4: We appreciate your helpful suggestion.

---

### Official Review · Reviewer_B5g3 · 2026-03-17

**Soundness:** 3
**Presentation:** 3
**Significance:** 3
**Originality:** 3
**Overall Recommendation:** 4
**Confidence:** 3

**Summary:**

The paper proposes a deterministic denoising algorithm for discrete diffusion models by replacing stochastic reverse sampling with a herding-based deterministic update rule. A notable domain studied by the article is discrete generative modeling for text, images, and combinatorial optimization tasks, where the approach improves generation quality and efficiency without retraining existing models. Overall, an important concept outlined by the paper is that deterministic dynamics can approximate categorical transition probabilities more faithfully than stochastic sampling, leading to better sample quality and faster convergence.

**Compliance With Llm Reviewing Policy:**

Affirmed.

**Final Justification:**

Although the method has not yet been compared against advanced higher-order discrete diffusion solvers, I belive the submission still has merits for publication at ICML.

**Key Questions For Authors:**

* The paper emphasizes heavily on deterministic sampling? What are the key advantages of them? Are there any downstream applications that are enabled via them?

* How does the proposed approach compare against more advanced discrete diffusion solvers, such as higher order solvers:  Fast Solvers for Discrete Diffusion Models: Theory and Applications of High-Order Algorithms https://arxiv.org/pdf/2502.00234

**Limitations:**

See above

**Strengths And Weaknesses:**

Strengths:

* This paper expands our understanding of uniform diffusion processes and proposes a new deterministic sampling procedure for discrete diffusion models based on a herding-inspired denoising mechanism.
 * The proposed process generates high-quality samples faster, as demonstrated across multiple experiments including text generation, image generation, and combinatorial optimization problems, outperforming commonly used Gumbel-Max sampling.
  * The paper studies the theoretical convergence properties of the proposed approach and shows that it enjoys a favorable convergence rate of $O(T^{-1})$, which is faster than standard stochastic sampling.
* Overall, the paper is of significant interest to the discrete diffusion community as it provides a simple yet effective alternative to stochastic reverse processes.

Weaknesses:
* The paper focuses primarily on uniform diffusion processes. While these models are generally considered less competitive than masked diffusion models in many generative modeling settings, studying them can still provide useful insights and help improve our understanding of discrete diffusion dynamics.
* The proposed method introduces an additional hyperparameter $\delta$ controlling the delayed switching mechanism, which appears to have a significant impact on performance and therefore must be tuned carefully.
* The introduction of $\delta$ requires stronger justification. In particular, the paper should better explain why this mechanism is necessary, how practitioners should choose its value in practice, and whether the theoretical guarantees (e.g., the $O(T^{-1})$ convergence rate) still hold under this modification.

---

> ### Author Rebuttal · Authors · 2026-03-31
>
> Thank you for your positive and constructive review. We appreciate your positive assessment of the novelty, significance, and practicality of our contribution. Below we respond point by point.
> ### Answers to Key Questions
> **(1) Advantages of deterministic sampling.**
> The main advantage of our proposed derandomization is improved efficiency and sample quality: it can achieve better quality for the same $T$, and comparable quality with smaller $T$, as demonstrated in our experiments. More broadly, deterministic denoising may also enable new downstream applications, analogous to what deterministic samplers have enabled in continuous diffusion models. For example, because the reverse process is deterministic given the initial state, one can imagine applications such as discrete morphing by interpolating between the initial states of two generated samples.
>
> **(2) Relation to advanced solvers.**
> Thank you for pointing out the relation to recent improvements of the reverse process. These approaches improve the approximation or design of the reverse Markov transition probabilities, whereas our approach derandomizes the Markov chain for given transition probabilities. Therefore, our algorithm is complementary to such methods and can in principle be combined with them. We agree that this is an important future direction.
>
> ### Clarifications on Weaknesses
> **(1) Uniform diffusion.**
> We agree with your point that masked diffusion models are studied extensively, especially in generative text modeling, and that nevertheless studying uniform diffusion remains important for improving our understanding of discrete diffusion dynamics. As a side note, the very recent works listed below have also suggested that uniform diffusion may have favorable scaling behavior compared with masked diffusion, which further motivates studying uniform diffusion. We also note that, as our approach derandomizes reverse Markov chains, it can in principle be applied to noising processes other than uniform diffusion.
> - von Rutte et al., Scaling Behavior of Discrete Diffusion Language Models, arXiv:2512.10858, ICLR 2026.
> - Deschenaux et al., The Diffusion Duality, Chapter II: $\Psi$-Samplers and Efficient Curriculum, arXiv:2602.21185, ICLR 2026.
> - Sahoo et al., Scaling Beyond Masked Diffusion Language Models, arXiv:2602.15014.
>
> **(2) Dependency on $\delta$.**
> Our experiments suggest that the dependence on $\delta$ is gradual and robust. While our manuscript reports near-best results, the proposed method improves over probabilistic Gumbel-max sampling for a broad range of $\delta$ values. For example, on LM1B, we observed that any $\delta \in [0, 0.5]$ improved generative PPL for both $T=128$ and $T=1024$ (see the table below).
>
> **(3) Delayed switching mechanism.**
> While a broad range of $\delta$ values improves performance, hyperparameter tuning is still important for achieving the best generative quality. As a practical guideline, we currently observe that larger $\delta$ tends to work better for smaller $T$. The theoretical guarantees in our paper, including the $O(T^{-1})$ discrepancy bound, also hold for $\delta>0$, since the proof of Proposition 1 incorporates the margin parameter $\delta$ into the lower bound. We agree, however, that the role of delayed switching should be studied further both theoretically and empirically.
>
> **Supplementary table.** Gen. PPL (entropy) of generated samples from the UDLM model on LM1B for various $\delta$ values.
>
> |          |       Gumbel |   $\delta=0$ | $\delta=0.0005$ | $\delta=0.005$ | $\delta=0.05$ | $\delta=0.5$ |
> | -------- | -----------: | -----------: | --------------: | -------------: | ------------: | -----------: |
> | $T=1024$ | 98.75 (6.83) | 46.26 (6.18) |    46.71 (6.30) |   56.02 (6.52) |  74.79 (6.70) | 94.18 (6.66) |
> | $T=128$  | 98.33 (6.81) | 75.67 (6.12) |    67.71 (6.15) |   60.39 (6.34) |  73.49 (6.66) | 97.21 (6.66) |

---

> > ### Author Rebuttal · Reviewer_B5g3 · 2026-04-07
> >
> > I would appreciate it if the proposed method could be compared against advanced discrete diffusion solvers, as pointed out in the original review.

---

> > > ### Author Response · Authors · 2026-04-07
> > >
> > > Thank you very much for this follow-up question. After the ICML submission, we trained a UDLM model on OpenWebText, which facilitates comparison with other methods. Below we compare our results with the advanced high-order solver mentioned in your review (Ren et al., arXiv:2502.00234, Table 3). Please note that this is not a direct comparison under the same model: their result combines the $\theta$-Trapezoidal solver with the masked diffusion model RADD, whereas ours combines herding-based denoising with the uniform diffusion model UDLM. That said, in terms of generative PPL, $\theta$-Trapezoidal performs better in the large-$T$ regime, while herding-based denoising performs better in the small-$T$ regime. This suggests that herding-based denoising is competitive with advanced solvers, and that it makes the uniform-diffusion approach more competitive as well. Furthermore, as noted previously, such inference-time techniques may in principle be combined with herding, which could lead to further improvements.
> > >
> > > | Method                         |     $T=16$ |    $T=32$ |    $T=64$ |   $T=128$ |   $T=256$ |   $T=512$ |  $T=1024$ |
> > > | ------------------------------ | -----: | ----: | ----: | ----: | ----: | ----: | ----: |
> > > | UDLM                           | 104.62 | 83.71 | 77.07 | 73.71 | 72.83 | 73.14 | 72.85 |
> > > | UDLM + herding ($\delta=0.05$) |  88.53 | 57.99 | 56.87 | 50.86 | 47.91 | 47.31 | 46.50 |
> > > | RADD + $\theta$-Trapezoidal    | 123.59 | 89.91 | 66.55 | 49.05 | 39.96 | 32.46 | 27.56 |

---

### Decision · Program_Chairs · 2026-04-30

**Decision:**

Reject

**Comment:**

The paper proposes a deterministic scheme for denoising diffusion models. The stochastic reverse sampling is replaced by a herding-based deterministic update rule.

The novelty of the contribution has been recognized. Useful theoretical results companions the idea, like boundedness of the weight vector.
Experimental evidence, albeit limited, is convincing: less steps are necessary for generation. The method also appears to be simple: it is a "drop-in" replacement of existing algorithm, making it fast to test. No re-training is necessary. On the other hand, the herding algorithm, while benefitting from some theoretical guarantees (good convergence rate), appears mostly heuristic.

Two reviewers (B5g3 and TrUM) found that the paper contained interesting ideas:
- "the paper is of significant interest to the discrete diffusion community as it provides a simple yet effective alternative to stochastic reverse processes"
- "I believe it could spark interest and generate further ideas for other readers if accepted"

Experiments are mostly small scale, which diminishes their impact. In language modeling, proper generative benchmarks are missing (e.g like GSM8k), which often demonstrate phase transitions when generative capabilities improve or degrade, when simple metrics like Mauve and Generative Perplexity move smoothly.

A significant effort was done by the authors during the rebuttal period, like training a new UDLM model on Openwebtext (discussion with Reviewer B5g3, TrUM). However, the scale is still too small and more complex downstream benchmarks are still missing. This concern was shared by Reviewer HYCN & GBXx.

Per authors abstract, since their method "serves as a direct replacement for the stochastic denoising process, without requiring retraining or continuous state embeddings" it makes it applicable to larger opensource models trained by the community. I suggest author focuses their efforts on scaling-up their idea on bigger LM and T2I models to strengthen their experimental section.
* A possible starting point are the checkpoints of the "Scaling Behavior of Discrete Diffusion Language Models" paper
 in https://huggingface.co/collections/dvruette/scaling-behavior-of-discrete-diffusion-language-models
* The work done in https://github.com/baaivision/URSA is another possibility to explore, to scale-up the vision part of the paper beyond Cifar-10

Based on the weakness of the experimental section, I recommend a rejection.